# Chemical Protein Crosslinking-Coupled Mass Spectrometry Reveals Interaction of LHCI with LHCII and LHCSR3 in *Chlamydomonas reinhardtii*

**DOI:** 10.3390/plants13121632

**Published:** 2024-06-13

**Authors:** Laura Mosebach, Shin-Ichiro Ozawa, Muhammad Younas, Huidan Xue, Martin Scholz, Yuichiro Takahashi, Michael Hippler

**Affiliations:** 1Institute of Plant Biology and Biotechnology, University of Münster, Schlossplatz 8, 48143 Münster, Germany; l_mose02@uni-muenster.de (L.M.); myounas@uni-muenster.de (M.Y.); martin.scholz@uni-muenster.de (M.S.); 2Institute of Plant Science and Resources, Okayama University, Kurashiki 710-0046, Japan; ozwsh1r@okayama-u.ac.jp; 3Research Institute for Interdisciplinary Science, Okayama University, Okayama 700-8530, Japan; taka@cc.okayama-u.ac.jp

**Keywords:** *Chlamydomonas reinhardtii*, light harvesting, state transitions, photoprotection, chemical crosslinking

## Abstract

The photosystem I (PSI) of the green alga *Chlamydomonas reinhardtii* associates with 10 light-harvesting proteins (LHCIs) to form the PSI-LHCI complex. In the context of state transitions, two LHCII trimers bind to the PSAL, PSAH and PSAO side of PSI to produce the PSI-LHCI-LHCII complex. In this work, we took advantage of chemical crosslinking of proteins in conjunction with mass spectrometry to identify protein–protein interactions between the light-harvesting proteins of PSI and PSII. We detected crosslinks suggesting the binding of LHCBM proteins to the LHCA1-PSAG side of PSI as well as protein–protein interactions of LHCSR3 with LHCA5 and LHCA3. Our data indicate that the binding of LHCII to PSI is more versatile than anticipated and imply that LHCSR3 might be involved in the regulation of excitation energy transfer to the PSI core via LHCA5/LHCA3.

## 1. Introduction

Oxygenic photosynthesis is based on a series of light-dependent reactions leading to water oxidation at the donor side of photosystem II (PSII), NADP^+^ reduction at the acceptor side of photosystem I (PSI) and ATP formation [1]. The ATP synthase produces ATP at the expense of the proton motive force that is generated by the light reactions [2]. The Cyt *b_6_f* complex establishes the link between the two photosystems by transferring electrons from the membrane bound plastoquinone (PQ) to a soluble carrier, plastocyanin (PC) or cytochrome c_6_ (Cyt c_6_). At the same time, the complex pumps protons from the chloroplast stroma into the thylakoid lumen. PC and Cyt c_6_ are oxidized by photo-oxidized PSI, whereupon PSI is able to photo-reduce ferredoxin (FDX). PSI is therefore a light-driven PC / Cyt c_6_ and FDX oxidoreductase. The core of PSI is highly conserved from cyanobacteria to vascular plants [3,4,5,6] and harbors approximately 100 chlorophyll molecules [4] serving as an antenna system that collects light energy. This core antenna is extended by additional chlorophyll-binding proteins that form the light-harvesting complex (LHCI). High-resolution structures from vascular plants revealed that PSI contains four LHCIs [7,8,9], whereas PSI of green algae may contain up to ten. Two antenna proteins, LHCA2 and LHCA9, bind between PSAG and PSAL and up to eight antenna proteins arrange in two crescents at the PSAF pole [10,11,12,13,14]. Recently, an additional LHCA1-LHCA4 dimer was found to be bound at the PSAL side in *Arabidopsis thaliana* [15], suggesting that this mode of organization is present in vascular plants as well. A trimeric chlorophyll-binding antenna protein complex, usually associated with PSII (LHCII), was also identified in this low-resolution complex in contact with the additional LHCA1-A4 dimer at the PSAL side.

The binding of LHCII to PSI in the context of state transitions occurs in most organisms of the green lineage [16] and balances the excitation energy between PSI and PSII [17,18]. In response to light conditions where PSII is preferentially excited, both PSII core and LHCII proteins are phosphorylated [19]. As a result, phosphorylated LHCII proteins detach from PSII and partially associate with PSI (state II). In response to conditions where PSI excitation predominates, this process is reversed. LHCII proteins are de-phosphorylated and bind to PSII (state I) [20,21]. The kinase responsible for LHCII phosphorylation is STT7 in *C. reinhardtii* [22] or STN7 in *A. thaliana* [23].

The structures of PSI-LHCI-LHCII complexes representing state II were revealed via high-resolution cryogenic electron microscopy (cryo-EM) from maize [24] and *C. reinhardtii* [25,26]. In these structures, phosphorylated LHCIIs were found in contact with peripheral PSI subunits, particularly PSAH, PSAL and PSAO, suggesting that the recognition pattern between the phosphorylated LHCII trimer 1 and the PSI core is conserved in the green lineage [25]. It is noteworthy that the PSI-LHCI-LHCII complex from *C. reinhardtii* contains two LHCII trimers [25]: LHCII trimer 2 associates with LHCA2 and PSAH via LHCBM5. LHCBM5 is phosphorylated and its phosphorylation depends on STT7 [27].

In a recent cryo-EM study, Naschberger et al. [28] identified a PSI-LHCI dimer from *C. reinhardtii* where two copies of LHCA9 tether two monomeric PSI-LHCIs in a head-to-head fashion, forming a large oligomeric protein complex. Notably, LHCA2 and PSAH are absent from this dimeric PSI-LHCI structure [28]. Cryo-EM analyses of PSI particles from a *C. reinhardtii* temperature-sensitive photoautotrophic PSII mutant (TSP4) also showed the presence of PSI-LHCI dimers [29]. Reversible PSI-LHCI dimerization may play a physiological role in thylakoid membrane structure maintenance. Importantly, the formation of PSI-LHCI-LHCII and PSI-LHCI dimerization are mutually exclusive, as PSI-LHCI dimer formation clashes with structural features of the reported state transition complex [28].

Another macromolecular PSI-LHCI organization state is the PSI-Cyt *b_6_f* supercomplex: Iwai et al. isolated a protein supercomplex from *C. reinhardtii* composed of PSI-LHCI, LHCII, Cyt *b_6_f* complex, FNR and PGRL1 (proton gradient regulation 5-like 1) from state II conditions [30]. In vitro spectroscopic analyses indicated that this supercomplex performed electron flow in the presence of exogenously added soluble PC and FDX [30]. Notably, Terashima et al. [31] isolated an in vitro active PSI-Cyt *b_6_f* supercomplex of similar composition from anaerobic growth conditions. In another work, a low-resolution structure of such a PSI-Cyt b_6_f supercomplex isolated from *C. reinhardtii* upon anaerobic incubation was obtained via low-resolution negative staining transmission electron microscopy [32]. In addition, Steinbeck et al. [32] identified trimeric LHCII complexes binding at the LHCA1-PSAG side, forming a new class of PSI-LHCI-LHCII complexes.

Energy-dependent non-photochemical quenching (qE-dependent NPQ) drives the thermal dissipation of excess excitation energy and thereby provides effective photo-protection in response to excess light. In vascular plants, PSBS, a putative PSII subunit, is mechanistically required for qE [33]. In *C. reinhardtii*, qE is mainly facilitated by LHCSR3 [34]. Among other factors, *LHCSR3* expression is induced in response to high light and low CO_2_ [34,35,36,37]. LHCSR3 is pH-responsive and converts to an energy-dissipative state in response to low pH conditions as shown by in vitro and in vivo experiments [38,39,40]. LHCSR3 binds to both PSII-LHCII and PSI-LHCI [35,41,42], with the molecular docking site at both photosystems remaining elusive. State transitions and qE-dependent NPQ have been suggested to play complementary roles in the high light response of *C. reinhardtii* [35].

In this work, we took advantage of chemical crosslinking of proteins in conjunction with mass spectrometry to identify protein–protein interactions between the light-harvesting proteins of PSI and PSII. We detected crosslinks suggesting the binding of LHCBM trimers to the LHCA1-PSAG side of PSI as well as protein–protein interactions of LHCSR3 with LHCA5 and LHCA3.

## 2. Results

The aim of this work was the identification of potential PSI-LHCI protein–protein interactions, as recently described [12], with a focus on high light conditions where state transitions and qE-dependent NPQ coincide. To this end, we used mass spectrometry to identify crosslinked peptides. Briefly, chemical protein crosslinking was performed with DSS (Disuccinimidyl suberate). DSS is a homobifunctional, non-cleavable and membrane-permeable crosslinker. It contains an amine-reactive *N*-hydroxysuccinimide (NHS) ester at each end of an 8-carbon spacer arm. Crosslinking was performed with thylakoid membranes isolated from the green alga *C. reinhardtii*. After crosslinking, membrane-associated protein complexes were solubilized with n-dodecyl α-D-maltoside (α-DDM), followed by separation with sucrose density gradient (SDG) centrifugation. The SDGs were fractionated and proteins present in individual fractions were separated by SDS-PAGE. The gel lanes were then cut into slices and subjected to in-gel digestion. The peptides were analyzed by liquid chromatography-coupled mass spectrometry (LC-MS/MS). For the analyses and identification of crosslinked peptides, four different algorithms were utilized. A total of 245 distinct crosslink combinations between different proteins were identified (inter-protein crosslinks, Appendix A). In this work, we focused on potential protein–protein interactions based on crosslinked peptides independently validated by at least two different algorithms or based on two distinct crosslinks between the same peptides. This subset contained eight crosslinks including at least one LHCA peptide (Table 1, Appendix A). Crosslinking between LHCA2-PSAB, LHCA2-PSAH, LHCA3-PSAK and LHCA4-LHCA8 have already been described [12] and structurally confirmed [10,28]. Additional crosslinks agreeing with the PSI-LHCI structure are LHCA3-LHCA5 and LHCA5-LHCA6. Surprisingly, we found several crosslinked peptides that suggest an interaction between LHCA1 and LHCBM proteins (Figure 1). For instance, the LHCA1 peptide SGEL[K]LK was found crosslinked to the peptides TV[K]PASK (LHCBM1), GTG[K]TAAK (LHCBM3) and TAA[K]QAPASSGIEFYGPNR (LHCBM3). In some cases, it was impossible to determine which of several candidate LHCBM proteins were crosslinked to LHCA1.This is due to the high sequence similarity of LHCBM proteins and the resulting occurrence of ambiguous peptides. For example, the same LHCA1 peptide mentioned above was also found crosslinked to the peptide AAAP[K]SSGVEFYGPNR, which could be derived from LHCBM4, LHCBM6, LHCBM8 or LHCBM9. Similarly, the LHCA1 peptide SGEL[K]LK was found crosslinked to the peptide TAA[K]AAAPK (LHCBM8, LHCBM6, LHCBM4). The fact that we observed crosslinking of the same LHCA1 peptide with peptides of several different LHCBM proteins substantiates the potential binding of LHCBM trimers to the LHCA1-PSAG side of PSI. Notably, the N-terminal threonine residues (T27) of the peptides T_27_V[K]PASK (LHCBM1) and T_27_AA[K]QAPASSGIEFYGPNR (LHCBM3) were found to be phosphorylated in the context of state transitions as demonstrated by mass spectrometry [43] and phosphorylated T27 of LHCBM1 was resolved in the latest cryo-EM structure of the PSI-LHCI-LHCII state transition complex [25]. As outlined above, two LHCII trimers were observed in the *C. reinhardtii* PSI-LHCI-LHCII complex, with trimer 1 consisting of LHCBM1, LHCBM2/7 and LHCBM3/4 and trimer 2 of LHCBM5, LHCBM2/7 and LHCBM3/4. In the crosslink between LHCA1 (SGEL[K]LK) and LHCBM4/LHCBM6/LHCBM8 (^22/23^TAA[K]AAAPK^30/31^) (Table 1), T22/T23 was also shown to be phosphorylated in state II [43]. Notably, crosslinking between LHCBM proteins and LHCA1 primarily occurred in the N-terminal region of these proteins. The loop containing the crosslinked lysine residues (K26 and K30) in LHCBM3 (Chain Y) of the LHCII trimer is not resolved in the currently available PDB structures. Therefore, to dock LHCII to PSI at the LHCA1-PSAG side based on the crosslinking data, we first predicted the structure of LHCBM3 using AlphaFold2 [44] (Appendix A), as two independent crosslinks between LHCBM3 and LHCA1 were found. The predicted LHCBM3 structure was then superimposed with LHCII trimer 1 (PDB ID: 7E0J; Appendix A) to obtain a final structure of LHCII trimer 1 with all the crosslinked residues being resolved (Appendix A). Subsequently, taking advantage of the crosslinking data and the potential positioning of LHCII trimers at the LHCA1-PSAG side previously observed in negatively stained particles [32], we modelled hypothetical structures of PSI-LHCI-LHCII with LHCII trimers binding at the LHCA1-PSAG side of PSI-LHCI (PDB ID: 7ZQC) (Figure 2 and Appendix A). In a first potential configuration (Figure 2A and Appendix A), the LHCII trimer is in contact with both LHCA1a and LHCA1b, incorporating the crosslinks observed between LHCBM3 and LHCA1 as well as LHCBM1 and LHCA1. The shortest distance between K26 of LHCBM3 (Chain Y) and K174 of LHCA1a is 31.5 Å, while the distance between K30 of LHCBM3 (Chain Y) and K174 of LHCA1a is 29.5 Å. Similarly, the distance between K29 of LHCBM1 (Chain Z) and K174 of LHCA1b is 27.4 Å (Figure 2A). All these distances are in the optimum range and satisfy the distance constraints of the DSS crosslinker [45,46]. In a second potential configuration (Figure 2B and Appendix A), the LHCII trimer is positioned at the outer LHCI belt, incorporating the crosslinks observed between LHCBM3 and LHCA1. Here, the shortest distances from K174 of LHCA1b to K26 and K30 of LHCBM3 (Chain Y) in LHCII are 32.5 and 26.6 Å, respectively (Figure 2B). Protein–protein interactions between various LHCBM proteins are illustrated by a large number of LHCBM inter-protein crosslinks (Appendix A).

Strikingly, we also observed protein crosslinking of LHCSR3 (peptide VMQT[K]ELNNGR) with LHCA5 (peptide ELQT[K]EIK) and LHCA3 (peptides EL[K]LKEIK and ELKL[K]EIK) (Table 1). The LHCSR3-LHCA5 crosslink was found independently by two algorithms, while regarding LHCSR3-LHCA3 two distinct crosslinks between the same peptides were detected by MaxQuant. The LHCA5 and LHCA3 peptides differ by only one amino acid and are located at corresponding positions within the protein structures of both LHCA proteins [10,28]. At the level of PSI-LHCI, LHCA5 and LHCA3 are adjacent to each other and located at the margin of the outer and inner LHCA belt, respectively. These findings imply that LHCA5 and LHCA3 represent potential binding sites of LHCSR3 at PSI-LHCI. Until now, no experimental structure of LHCSR3 has been resolved, so we first modelled the structure of LHCSR3 with AlphaFold2 [44] using the sequence retrieved from the UniProt database (Appendix A). The transit peptide was predicted with TargetP-2.0 [48,49] and the first 32 amino acids were removed before modelling the structure with AlphaFold2. As expected, the model located the pH-responsive C-terminal residues to the thylakoid lumen [38,40,50]. Moreover, it is noteworthy that the structure predicted for LHCSR3 closely resembled that of the LHCA proteins, as exemplified by a superposition of the predicted model with LHCA8 (PDB ID: 7ZQC) from *C. reinhardtii* (Appendix A). The same residue of LHCSR3 (K196) was observed crosslinked to two different LHCA proteins (K185 of LHCA5 and K214 as well as K216 of LHCA3). In a hypothetical model, LHCSR3 was docked to PSI-LHCI (PDB ID: 7ZQC) in a position that allows for the occurrence of all crosslinks based on a single binding site of LHCSR3 at LHCA5/LHCA3 (Figure 3). The structural similarity of LHCSR3 with the LHCA proteins was also reflected in the orientation of the transmembrane helices in the hypothetical model (Figure 3A). The distance between K196 of LHCSR3 and K185 of LHCA5 as well as K196 of LHCSR3 and K216 of LHCA3 is 22.7 Å each, whereas the distance from K196 of LHCSR3 to K214 of LHCA3 is 30.7 Å in the predicted PSI-LHCI-LHCSR3 complex (Figure 3B). All these distances are in the optimum range and satisfy the distance constraints of the DSS crosslinker [45,46].

Depletion of LHCA3 destabilizes the entire PSI-LHCI complex [51]. Therefore, we took advantage of a *lhca5* insertional mutant [12] (see Appendix A for the characterization of the genomic insertion). In this mutant, the abundance of LHCA5 was strongly diminished at the level of whole cells [12]. Moreover, the amounts of LHCA1, LHCA6 and LHCA4 in the outer belt as well as LHCA3 in the inner belt were lower at the level of isolated PSI-LHCI [12]. The *lhca5* insertional mutant was complemented with the *LHCA5* wild type gene, which restored LHCA5 amounts (Appendix A). To investigate the binding of LHCSR3 to PSI-LHCI, we shifted the *lhca5* insertional mutant along with the complemented strain to low CO_2_ (medium without carbon source) and high light (150 µmol m^−2^ s^−1^) to induce expression of *LHCSR3*. Thylakoid membranes were isolated and solubilized with α-DDM. Subsequently, SDG centrifugation was performed. Six green bands were observed after centrifugation, which were designated from top to bottom as LHCII monomer, LHCII trimer, PSII core, PSI-LHCI, PSI-LHCI-LHCII and PSII-LHCII (Figure 4A, Appendix A). PSI-LHCI and PSI-LHCI-LHCII particles from the LHCA5 complemented strain migrated to zones of higher sucrose densities as compared to the particles from the *lhca5* insertional mutant, indicating that, as observed previously [12], the absence of LHCA5 likely destabilized other LHCA subunits, accordingly lowering the density of the respective PSI-LHCI particles (Figure 4A, Appendix A). In the next step, SDG fractions containing PSII-LHCII and PSI-LHCI complexes (Figure 4B) were separated via SDS-PAGE, followed by immunoblotting (Figure 4C, Appendix A). Based on the detection of PSAA and PSBA as well as LHCBM1 and LHCA3, LHCA2 and LHCA9, migration patterns of PSII-LHCII, PSI-LHCI-LHCII and PSI-LHCI complexes were defined (Figure 4C, Appendix A). As observed previously, PSII-LHCII complexes migrated to the highest sucrose density, followed by PSI-LHCI-LHCII and PSI-LHCI. As expected, the abundance of LHCA5 was strongly diminished in the *lhca5* insertional mutant, whereas it was restored in the LHCA5 complemented strain (Figure 4C, Appendix A). Notably, LHCSR3 was equally present in PSII-LHCII, PSI-LHCI-LHCII and PSI-LHCI fractions in the LHCA5 complemented strain. In the *lhca5* insertional mutant however, the association of LHCSR3 was observed in PSII-LHCII and PSI-LHCI-LHCII but only faintly in PSI-LHCI fractions. This trend was confirmed by several independent experiments, where LHCSR3 binding was more pronounced in the PSI-LHCI fraction of the LHCA5 complemented strain than in the PSI-LHCI fraction of the *lhca5* insertional mutant (Figure 4C, Appendix A), while LHCSR3 binding to PSI-LHCI-LHCII occurred in both strains.

## 3. Discussion

Chemical protein crosslinking in conjunction with mass spectrometry revealed binding of LHCBM proteins to the LHCA1-PSAG side of PSI-LHCI and binding of LHCSR3 to LHCA3 and LHCA5 at the inner and outer LHCA belt at the PSAF pole.

Crosslinks between LHCA1 and N-terminal residues of LHCBM proteins always involved the same LHCA1 peptide. Two proteotypic peptides of LHCBM3 and one proteotypic peptide of LHCBM1 were found crosslinked with LHCA1, identifying LHCMB3 and LHCBM1 as potential binding partners of LHCA1. Furthermore, non-proteotypic peptides of LHCBM4, LHCBM8, LHCBM6, LHCBM9 and LHCBM1 were detected as crosslinking partners of the LHCA1 peptide. In this case, one or several of these proteins may be a binding partner of LHCA1. Based on our data, we are currently unable to distinguish between the binding of individual LHCBM proteins and the binding of LHCII trimers to LHCA1. Yet, binding of LHCII trimers to the LHCA1-PSAG side of PSI-LHCI has already been observed via low-resolution negative staining transmission electron microscopy of PSI particles isolated from *C. reinhardtii* upon anoxic incubation [32]. A hypothetical model based on the interaction between LHCBM3 and LHCA1 peptides visualized the potential association of an LHCII trimer with the inner and outer LHCA1 proteins (Figure 2 and Appendix A). During the process of state transitions, two LHCII trimers may bind to the PSAH-PSAL-PSAO side of PSI-LHCI in *C. reinhardtii* [25,26]. Although we observed crosslinking of PSAH with LHCA2, we failed to detect crosslinking between PSAH and LHCBM1. This may be due to the absence of residues in a favorable position and/or insufficient accessibility for DSS required for chemical crosslinking followed by LC-MS/MS detection at the interface between PSAH and LHCBM1. In *A. thaliana*, it was shown that LHCI facilitates excitation energy transfer of loosely bound “additional” LHCII proteins to the PSI core [52,53,54,55]. Yadav et al. 2017 [56] also observed binding of LHCII trimers at LHCI via low-resolution negative staining transmission electron microscopy. However, the molecular binding site of these additional LHCII proteins at LHCI is elusive. Based on our findings, we propose that binding of additional LHCII monomers and/or trimers mediating excitation energy transfer to the PSI core antenna may occur via LHCA1 in *C. reinhardtii* and possibly also in vascular plants. Notably, N-terminal LHCBM peptides were found to be crosslinked with LHCA1, including T27 of LHCBM1, which is phosphorylated during state transitions [25]). However, there is no evidence at this point that phosphorylation modulates binding of LHCBM proteins to LHCA1. In fact, we observed crosslinks between non-phosphorylated LHCBM and LHCA1, suggesting phosphorylation is not required for LHCBM binding to LHCA1.

Moreover, we found crosslinking of LHCSR3 with LHCA5 and LHCA3 (Figure 3). Crosslinking occurred between the C-terminal LHCSR3 peptide VMQTKELNNGR and LHCA5 (peptide ELQTKEIK) and LHCA3 (peptide EL[K]LKEIK and peptide ELKL[K]EIK), close to the STT7-independent phosphorylation site S258 of LHCSR3 [27]. Whether C-terminal LHCSR3 phosphorylation is involved in LHCSR3 binding to LHCA5 and LHCA3 is currently unclear.

It has been revealed that LHCA3 is an important entry point for excitation energy transfer to the PSI core antenna in vascular plants and green algae [9,10,13]. Moreover, LHCSR3 has been shown to facilitate excitation energy quenching at PSI-LHCI in response to high light [57,58]. It has also been suggested that excitation energy transfer from the outer to the inner LHCA belt occurs via pigments in LHCA5-LHCA3-PSAA [10,13]. Thus, binding of LHCSR3 to LHCA5/LHCA3 may provide an additional level of regulation modulating excitation energy flux at PSI-LHCI in *C. reinhardtii*. Analyses of a *lhca5* insertional mutant supported the notion that LHCA5 is required for efficient binding of LHCSR3 to PSI-LHCI (Figure 4 and Appendix A). However, our data also revealed comigration of LHCSR3 with LHCII proteins bound to PSI-LHCI independently of direct binding to PSI-LHCI. In conclusion, it is possible that LHCSR3 binds to PSI-LHCI both directly via LHCA5/LHCA3 as well as indirectly via association with LHCII proteins to regulate excitation energy transfer to the PSI core during the process of state transitions.

## 4. Materials and Methods

### 4.1. Strains and Growth Conditions

Experiments were performed using the wild-type strain 4a+ [34], a *lhca5* insertional mutant (LMJ.RY0402.044057) [12] and a corresponding complemented strain. Cells were maintained at 25 °C in the presence of continuous light (20 µmol photons m^−2^ s^−1^) on TAP medium, solidified with 1.5% w/v agar. For experimental analyses, cells were cultured in TAP medium on a rotary shaker (120 rpm) at 25 °C in the presence of continuous light (20 µmol photons m^−2^ s^−1^) until the early mid-log phase. Prior to thylakoid isolations, cells were set to 4 µg mL^−1^ chl and shifted to photoautrophic conditions (medium without carbon source) for 40 h and high light (150 µmol m^−2^ s^−1^) for 24 h.

### 4.2. Generation of the LHCA5 Complemented Strain

The *LHCA5* gene (Cre10.g425900.t1.1, version 5.6) was amplified from cc125 *C*. *reinhardtii* genomic DNA. The primers for amplification annealed 2488 bp upstream and 886 bp downstream of the LHCA5 transcript. The amplified DNA was fused at the EcoRI site of the vector pBSSK by InFusion to obtain the p*ga5* plasmid. The p*ga5* plasmid was linearized by digesting with PsiI and transformed into the *lhca5* insertional mutant by electroporation with the NEPA21 (Neppagene, Ichikawa, Japan) according to [59]. Transformants were selected on TAP plates supplemented with 50 μg hygromycin under 50 μmol photons·m^−2^·s^−1^. For the PCR of *LHCA5* gene cloning, the KOD FX Neo (TOYOBO, Osaka, Japan) was used. The Tks Gflex polymerase (TAKARA, Kusatsu, Japan) was used for genotyping.

The following primers were used to amplify around the CIB1 cassette insertion site in the *lhca5* insertional mutant (LMJ.RY0402.044057). These primers were already listed in the CLiP library database. Lhca5_SGP1 and Lhca5_SGP2 were utilized to distinguish the recipient *lhca5* from the clones carrying the DNA fragment containing the *LHCA5* gene.

Lhca5_SGP1  CTACAAGAACTTCGGCTCGG

Lhca5_SGP2  CGGTCAACAATCGAGGTTTT

In the subsequent DNA sequencing of the PCR product, CIB 5′ and CIB 3′ were used. These primers were also listed in the CLiP library database.

CIB_5′  GCACCAATCATGTCAAGCCT

CIB_3′  GACGTTACAGCACACCCTTG

The following primers were used to amplify the DNA fragment containing the *LHCA5* gene (Cre10.g425900.t1.1, version 5.6).

If-EcoRI_ga5_F-up  GCTTGATATCGAATTGCATCCCCACATTCAAGAGC

If-EcoRI_ga5_R-up  CGGGCTGCAGGAATTGTGGGCGTGCGGGTTCGTTT

Up_a5_F  CGCCATCATTTGCACTTTCAAGCCCTCCCTCGCCG

Dwn_ga5_R  AGGACGCTGGCAAGAAGGACGCTGCCAAGAAGGAC

The following primers were used to check for PCR errors in the amplified DNA fragment in the plasmid vector.

seq_pga5_1_F  CCGGTGCATACGGCAAGGTTCGGTCGGGGGCTGGG

seq_pga5_1_R  CCCAGCCCCCGACCGAACCTTGCCGTATGCACCGG

seq_pga5_2_F  GTCCCTGGGGGTCCGTCGGGTTTCGGGAGCCCATC

seq_pga5_3_F  CCAAACCTTAAAAACGCCACTCATGTCAGAGGCTC

seq_pga5_4_F  AGCTCCCTGCCTTTGTAACTTTAACTCTGGTGCTT

seq_pga5_5_F  GTACCGCCAGTCGGAGCTGCAGCACGCTCGCTGGG

seq_pga5_6_F  AGACCAAGGAGATCAAGAACGGCCGCCTGGCCATG

seq_pga5_7_F  CATCCCCCTGACCTGCCTGTGGCCCGGCAGCCAGT

seq_pga5_8_F  ACGGGTGTGTAACCCAAAACCTCGATTGTTGACCG

### 4.3. Thylakoid Isolation and Chemical Crosslinking

Thylakoid isolations were performed according to Chua et al. [60] with modifications. All following steps were performed at 4 °C and dim light. Cells were disrupted in 0.33 M sucrose, 25 mM HEPES–KOH pH 7.5, 5 mM MgCl_2_, 1 mM PMSF,1 mM benzamidine and 5 mM aminocaproic acid with a nebulizer (2 bar, two passages). Broken cells were centrifuged at 32,800× *g* for 10 min (Beckman Coulter, Brea, California, US, JA-25.50 rotor, 20,000 rpm). The pellet was carefully resuspended in 0.5 M sucrose, 5 mM HEPES–KOH pH 7.5, 10 mM EDTA,1 mM benzamidine and 5 mM aminocaproic acid with a potter homogenizer. The resuspended material was layered on top of a sucrose density step gradient (1.8 M and 1.3 M sucrose, 5 mM HEPES–KOH pH 7.5, 10 mM EDTA, 1 mM benzamidine and 5 mM aminocaproic acid). Thylakoid membranes were extracted via ultracentrifugation at 70,800× *g* for 1 h and 20 min (Beckman Coulter SW 32 Ti rotor, 24,000 rpm). Thylakoids were collected from the step gradient interphases with a Pasteur pipet, diluted four times with 5 mM HEPES–KOH pH 7.5 and centrifuged at 37,900× *g* for 20 min (Beckman Coulter JA 25.50 rotor, 21,500 rpm).

For chemical crosslinking, thylakoids isolated from the wild-type strain 4a+ were set to 1.5 mg chl mL^−1^ in 5 mM HEPES–KOH pH 7.5. Chemical protein crosslinking was performed by adding 5 µL of 50 mM disuccinimidyl suberate (DSS; DSS-H12 and DSS-D12 at equimolar ratio, Creative Molecules) in DMSO to a thylakoid suspension corresponding to 400 µg chl. The crosslinking reaction was incubated for 30 min at room temperature in the dark and quenched by the addition of 400 mM Tris–HCl pH 7.5 to a final concentration of 50 mM.

### 4.4. Thylakoid Solubilization and Sucrose Density Gradient Centrifugation

The solubilization of thylakoid membrane proteins and the fractionation of photosynthetic protein complexes were performed according to Tokutsu et al. [61] with modifications. 10% n-Dodecyl α-D-maltoside (α-DDM) was added to crosslinked samples to a final concentration of 0.9% and samples were incubated at 4 °C for 5 min. Thylakoids isolated from the *lhca5* insertional mutant and the corresponding complemented strain were set to 1 mg chlorophyll mL^−1^ in 5 mM HEPES–KOH pH 7.5 and solubilized by addition of an equal volume of 2% α-DDM for 10 min. Unsolubilized material was separated by centrifugation.

Solubilized thylakoids (~250 μg chlorophyll) were loaded onto a 1.3 M to 0.1 M sucrose density gradient (SDG) including 5 mM HEPES–KOH pH 7.5 and 0.02% α-DDM. SDG fractions including PSII-LHCII, PSI-LHCI and PSI-LHCI-LHCII were collected after ultracentrifugation at 134,400× *g* (Beckman Coulter SW 41 Ti rotor, 33,000 rpm) for 14–22 h for further analysis.

### 4.5. SDS-PAGE

SDG fractions were supplemented with loading buffer and incubated at 65 °C for 15 min. Samples were loaded based on equal volume. Proteins were separated by 13% (*w*/*v*) SDS–PAGE [62]. Gels were stained with Coomassie Brilliant Blue R-250 and each lane was cut horizontally into 25 equal-sized gel slices, which were transferred to Eppendorf tubes for storage at −20 °C until in-gel protein digestion. Alternatively, gels were blotted onto nitrocellulose membranes (Cytiva, Amersham, UK) for detection with specific primary antibodies.

### 4.6. Immunoblotting

Immunoblotting was performed with primary antibodies against PSAA, PSBA (Agrisera, Vännäs, Sweden), LHCBM1, LHCSR3 [63], LHCA3 [64], LHCA5, LHCA2 and LHCA9 [12] and PSAD [51]. The antibody against PSAA was raised using the peptides STPEREAKKVKIAVDR and VKIAVDRNPVETSFEK and was obtained from Kaneka Eurogentec, Seraing, Belgium. The antibody against LHCBM1 was raised using the peptide CGAFTGEPPSY. All primary antibodies were used at a 1:1,000 dilution, except anti-PsbA (1:10,000). Secondary antibodies for ECL detection were used at a 1:10,000 dilution (goat anti-rabbit IgG (H + L)-HRP conjugate, Bio-Rad, Hercules, CA, USA).

### 4.7. In-Gel Digestion

Gel slices were subjected to tryptic in-gel digestion according to established protocols [65]. The resulting peptides were desalted using in-house made Stage-Tips packed with C18 solid phase extraction membranes (Supelco, Bellefonte, PA, USA) [66]. The desalted peptides were dried by vacuum centrifugation and stored at −80 °C until further use.

### 4.8. LC-MS/MS Analysis

Crosslinked peptide samples were analyzed on an LC-MS/MS system consisting of an Ultimate 3000 nano HPLC system (Thermo Fisher Scientific, Waltham, MA, USA) coupled via an ESI interface (Nanospray Flex, Thermo Fisher Scientific) to a Q Exactive Plus mass spectrometer (Thermo Fisher Scientific). Samples were resuspended in solvent A1 (0.05% trifluoroacetic acid (TFA)/4% acetonitrile (AcN)/ultrapure water) and loaded on a trap column (C18 PepMap 100, 300 µM × 5 mm, 5 µm particle size, 100 Å pore size; Thermo Fisher Scientific) at a flow rate of 7.5 µL min^−1^ for 5 min using solvent A1. Peptides were eluted in backflush mode from the trap column onto the separation column (Acclaim PepMap100 C18, 75 µm × 15 cm, 3 µM particle size, 100 Å pore size, Thermo Scientific). Flow rate was 300 nl min^−1^. The eluents for peptide separation were 0.1% formic acid in ultrapure water (A2) and 80% AcN/0.1% formic acid in ultrapure water (B). The following gradient was applied for elution: 2.5% B to 7.5% B over 4 min, 7.5% B to 40% B over 24 min, 40% B for 3 min, 40% B to 99% B over 3 min and 99% B for 10 min.

MS full scans (*m*/*z* 400-2000) were recorded at a resolution of 70,000 (FWHM at 200 *m*/*z*) with internal lock mass calibration on *m*/*z* 445.12003. Fragmentation spectra (MS2) were acquired at a resolution of 35,000 (FWHM at 200 *m*/*z*) and in a data-dependent way, where the 12 most intense ions from a full scan were fragmented by higher-energy c-trap dissociation (HCD) at a normalized collision energy of 27 and an isolation window of 2 *m*/*z*. The automatic gain control (AGC) targets for MS full scans (MS1) and MS2 were 1e6 and 1e5, respectively. The intensity threshold for MS2 was 1e4. The maximum injection times were set to 50 ms for MS1 and 100 ms for MS2. Ions with unassigned charge states or charge states 1 and 2 and 8 or above were excluded from fragmentation. Each sample was analysed twice, once with and once without the mass tags option enabled. Mass tags were used to achieve preferred fragmentation of crosslinked peptides, which were characterized by ion pairs with a mass difference of 12.07573 Da, resulting from the combined use of light (DSS-H12) and heavy (DSS-D12) crosslinkers. 

### 4.9. MS Data Analysis

MS raw files were searched against polypeptide sequences of selected *C. reinhardtii* proteins involved in photosynthetic light reactions (PSAx, PSBx, PETx, LHCIx, LHCIIx) Four search engines were employed to identify peptides crosslinked by DSS-H12 and DSS-D12: XiSearch 1.7.6.7 [67], MaxQuant 2.0.3.0 [68], Merox 2.0.1.4 [69] and Mass Spec Studio [70]. For crosslink identification by XiSearch and Merox, spectral files were converted to mgf format using MSConvert 3.0.23326 [71]. Precursor and fragment ion mass accuracies were 10 and 15 ppm, respectively. Minimum peptide length was 6. Acetylation of N-termini and oxidation of methionine were set as variable modifications. Crosslinked peptides were filtered to achieve a false discovery rate (FDR) of 0.05. Due to the fact that Mass Spec Studio did not calculate correct FDRs, leading to a high number of false positive identifications (decoy hits) among crosslinked peptides, the correct FDR was calculated based on e-values according to the target-decoy approach as described by Elias and Gygi [72]. Detailed descriptions of search settings are available in Supplemental Document S1.

### 4.10. Structural Modelling

For the structure prediction of LHCBM3 (Chain Y) and LHCSR3, AlphaFold2 [44] was employed, using sequences retrieved from the UniProt database. Before modelling the structure with AlphaFold2, the transit peptide of LHCSR3 was predicted with TargetP-2.0, while LHCBM3 was modelled in a way that the N-terminal crosslinked residues determined by mass spectrometry were included. The best models were selected from the list of generated models based on the AlphaFold confidence score (pLDDT). The predicted models were manually docked to PSI-LHCI using PyMOL (The PyMOL Molecular Graphics System, Version 2.4.1 Schrödinger, LLC, New York, NY, USA). The PSI-LHCI structure used for docking was retrieved from the PDB database (PDB ID: 7ZQC). The final figures were generated using PyMOL and UCSF ChimeraX [73,74].

## Figures and Tables

**Figure 1 plants-13-01632-f001:**
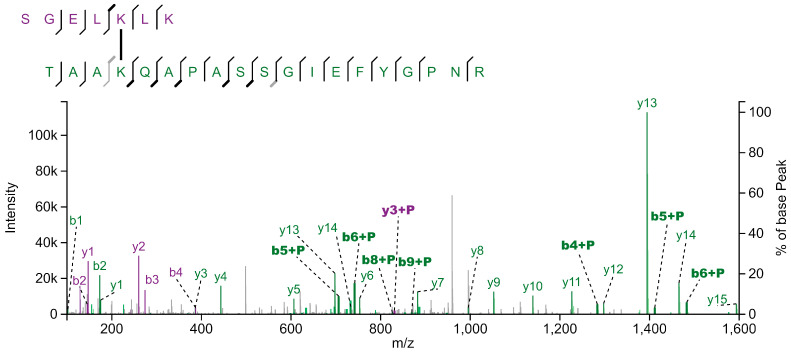
Fragmentation spectrum of crosslinked peptides from LHCA1 (SGELKLK) and LHCBM3 (TAAKQAPASSGIEFYGPNR). Isolated thylakoid membranes were crosslinked using DSS and subsequently solubilized with α-DDM. Photosynthetic protein complexes were separated by sucrose density gradient centrifugation, followed by SDS-PAGE of SDG fractions. After tryptic in-gel digestion of gel bands, peptide samples were submitted to LC-MS/MS analysis. Fragment ions originating from the LHCA1 and LHCBM3 peptides are shown in purple and green, respectively. Ions of those fragments that carry the cross-linked peptide are labelled with "+P". Spectrum annotation with XiView [47].

**Figure 2 plants-13-01632-f002:**
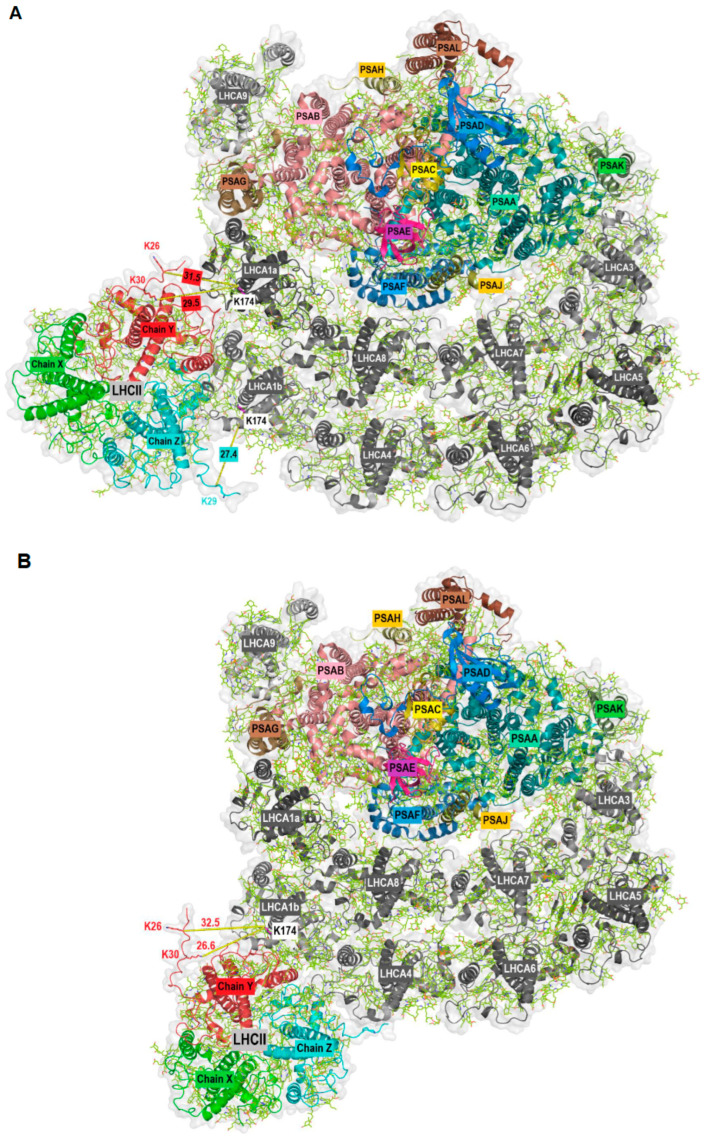
Stromal view of the docking of LHCII (PDB ID: 7E0J with newly modelled Chain Y) to PSI-LHCI (PDB ID: 7ZQC) at two potential positions (**A**,**B**) based on crosslinking data and the density observed in negatively stained particles [32]. PSI core and LHCII subunits are in color, LHCI are in grey. Chain X: LHCBM2/7, Chain Y: LHCBM3/4, Chain Z: LHCBM1. The shortest distance between the crosslinked lysine residues is indicated.

**Figure 3 plants-13-01632-f003:**
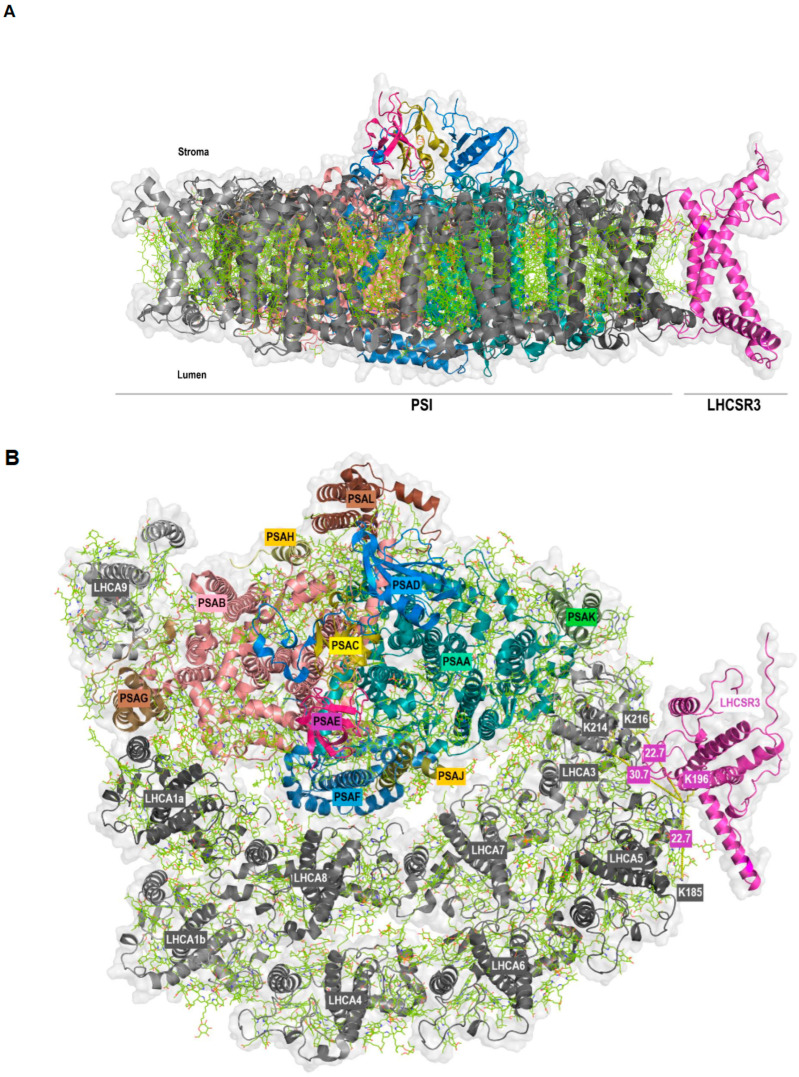
Membrane (**A**) and stromal (**B**) view of the docking of LHCSR3 to PSI-LHCI (PDB ID: 7ZQC) based on crosslinking data. PSI core subunits and LHCSR3 are in color, LHCI are in grey. The shortest distance between the crosslinked lysine residues is indicated (**B**).

**Figure 4 plants-13-01632-f004:**
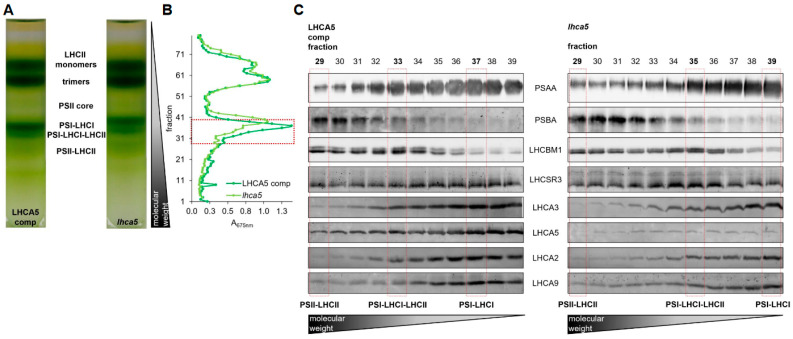
(**A**) Sucrose density gradients of α-DDM solubilized thylakoids (~250 µg chl) isolated from the *lhca5* complemented strain and the *lhca5* insertional mutant. Cells were shifted to low CO_2_ (medium without carbon source) for 40 h and high light (150 µmol m^−2^ s^−1^) for 24 h prior to the isolation. (**B**) Absorption profile of SDG fractions at 675 nm. SDG fractions selected for SDS-PAGE followed by immunoblotting are indicated by the red box. (**C**) Immunoblots against PSAA, PSBA, LHCBM1, LHCSR3, LHCA3, LHCA5, LHCA2 and LHCA9. Samples were loaded on equal volume. Peak fractions of PSII-LHCII, PSI-LHCI-LHCII and PSI-LHCI are indicated by red boxes.

**Table 1 plants-13-01632-t001:** Potential protein–protein interactions involving at least one LHCA protein based on crosslinked peptides independently validated by at least two algorithms or based on two distinct crosslinks between the same peptides. MSS = Mass Spec Studio, MQ = MaxQuant, Mx = Merox, Xi = XiSearch.

Protein 1 Uniprot ID	Protein 1 Name	Protein 1 PhytozomeID	Protein 2 Uniprot ID	Protein 2 Name	Protein 2 Phytozome ID	Peptide 1	Protein Position 1	Peptide 2	Protein Position 2	Search Engines
Q05093	LHCA1	Cre06.g283050	Q8S3T9, A8J270, A8J287, A8J264	LHCBM9, LHCBM8, LHCBM6, LHCBM4	Cre06.g285250, Cre06.g284200, Cre06.g283950, Cre06.g284250	SGELKLK	K174	TAAKAAAPK	K25	Xi; Mx; MQ
Q05093	LHCA1	Cre06.g283050	A8J264, A8J270, A8J287, Q8S3T9	LHCBM4, LHCBM8, LHCBM6, LHCBM9	Cre06.g283950, Cre06.g284250, Cre06.g285250, Cre06.g284200	SGELKLK	K174	AAAPKSSGVEFYGPNR	K31	MQ; Xi
Q05093	LHCA1	Cre06.g283050	Q93VE0	LHCBM1	Cre01.g066917	SGELKLK	K174	TVKPASK	K29	MSS; MQ; Xi
Q05093	LHCA1	Cre06.g283050	Q93WL4	LHCBM3	Cre04.g232104	SGELKLK	K174	GTGKTAAK	K26	Xi; MQ
Q05093	LHCA1	Cre06.g283050	Q93WL4	LHCBM3	Cre04.g232104	SGELKLK	K174	TAAKQAPASSGIEFYGPNR	K30	Xi; MQ
Q05093, Q75VZ0, A8IKC8, Q75VY9	LHCA1, LHCA4, LHCA2, LHCA3	Cre06.g283050, Cre10.g452050, Cre12.g508750, Cre11.g467573	Q42687	ATPD	Cre11.g467569	LKELKNGR	K179	LSALIMNPVVESDKK	K89	Xi; MQ
A8IKC8	LHCA2	Cre12.g508750	P13352	PSAH	Cre07.g330250	RYEIYKK	K130	YPDNQAKFFTQATDIISR	K68	Xi; MSS; MQ
A8IKC8	LHCA2	Cre12.g508750	P13352	PSAH	Cre07.g330250	YEIYKK	K130	YPDNQAKFFTQATDIISR	K68	Xi; MSS; MQ
A8IKC8	LHCA2	Cre12.g508750	P13352	PSAH	Cre07.g330250	KTGETGFLSFAPFDPMGMK	K131	YPDNQAKFFTQATDIISR	K68	MQ; Xi
A8IKC8	LHCA2	Cre12.g508750	P09144	PSAB	CreCp.g802312	TLNPGKESVPYFPWNEPWNKV	K231	LFPKFSQGLAQDPTTR	K8	MSS; MQ; Xi
A8IKC8	LHCA2	Cre12.g508750	P09144	PSAB	CreCp.g802312	ESVPYFPWNEPWNKV	K245	LFPKFSQGLAQDPTTR	K8	Mx; Xi; MSS; MQ
Q75VY9	LHCA3	Cre11.g467573	Q75VY8	LHCA5	Cre10.g425900	ELKLKEIK	K216	VPNPEMGYPGGIFDPFGFSKGNLK	K176	MQ; Xi
Q75VY9	LHCA3	Cre11.g467573	P93663	LHCSR3.2	Cre08.g367400	ELKLKEIK	K214	VMQTKELNNGR	K196	MQ
Q75VY9	LHCA3	Cre11.g467573	P93663	LHCSR3.2	Cre08.g367400	ELKLKEIK	K216	VMQTKELNNGR	K196	MQ
Q75VY9	LHCA3	Cre11.g467573	P14225	PSAK	Cre17.g724300	SIAKVDR	K37	FGLAPTVKK	K58	MSS; MQ; Xi
Q75VY9	LHCA3	Cre11.g467573	P14225	PSAK	Cre17.g724300	SIAKVDR	K37	NTTAGLKLVDSK	K66	MSS; MQ; Xi
Q75VZ0	LHCA4	Cre10.g452050	Q75VY7	LHCA8	Cre06.g272650	LKWYAQAELMNAR	K94	EADKWADWK	K184	Xi; MSS; Mx; MQ
Q75VY8	LHCA5	Cre10.g425900	Q75VY6	LHCA6	Cre06.g278213	GNKVPNPEMGYPGGIFDPFGFSK	K156	ASSRPLWLPGSTPPAHLK	S28	Xi; Mx
Q75VY8	LHCA5	Cre10.g425900	P93663	LHCSR3.2	Cre08.g367400	ELQTKEIK	K185	VMQTKELNNGR	K196	MQ; Xi

## Data Availability

The mass spectrometry proteomics data were deposited to the ProteomeXchange Consortium via the PRIDE [75] partner repository with the dataset identifier PXD051493. Dynamic exclusion was enabled with an exclusion duration of 30 s and a tolerance of 5 ppm.

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
