# Peer review of "Chemical Protein Crosslinking-Coupled Mass Spectrometry Reveals Interaction of LHCI with LHCII and LHCSR3 in Chlamydomonas reinhardtii"

_plants, 2024, doi:10.3390/plants13121632_

Round 1
Reviewer 1 Report
Comments and Suggestions for Authors
The work entitled “Chemical Protein Crosslinking-Coupled Mass Spectrometry Reveals Interaction of LHCI with LHCII and LHCSR3 in C. reinhardtii” suggests the existence of specific protein-protein interactions between PSII- and PSI-specific antenna proteins. The work also highlights the involvement of LHCSR3 in binding PSI via LHCA3 and LHCA5. Overall, the work is very well written and informative. It is also very clear in the description of the results and precise in the discussion section where all the relevant literature has been comprehensively cited. I have a few comments that in my opinion should be addressed though, as follows:
Line 15: in my opinion the word “controlled” does not fit the meaning of the sentence.
Lines 20-21: in this work the authors didn’t demonstrate the role of LHCSR3 in the regulation of the excitation energy transfer to PSI. I suggest toning down the sentence.
Lines 82-83: I wouldn’t refer to luminal acidification as a process of acclimation. In my opinion, lumenal acidification is faster than acclimation, which calls also for proteins and pigments synthesis indeed.
Line 93: data within this work suggest the existence of specific protein-protein interactions but this work didn’t demonstrate their existence.
Lines 101-102: what’s the effect of crosslinking on the biological sample? Can it affect protein integrity or generate technical biases? I think this topic deserves a few sentences in the discussion section.
Lines 115-116: How many peptides are needed before an interaction can be claimed with sufficient confidence?
Lines 176-177: How many algorithms are needed before an interaction can be claimed with sufficient confidence?
Lines 214-215: I suggest being more specific in the description of the environmental conditions.
Lines 313-316: What’s the expected impact of such light environmental conditions during the growth of the biological material on the conclusions drawn within this work? I would suggest including a brief discussion on the topic.
Lines 371-375: Crosslinking was performed after thylakoids isolation. Is this practice expected to affect the reliability of the results?
Lines 382-384: It seems the thylakoids of the mutant were treated differently. Is that so? Why? This needs clarification.
Author Response
Reviewer 1
The work entitled “Chemical Protein Crosslinking-Coupled Mass Spectrometry Reveals Interaction of LHCI with LHCII and LHCSR3 in C. reinhardtii” suggests the existence of specific protein-protein interactions between PSII- and PSI-specific antenna proteins. The work also highlights the involvement of LHCSR3 in binding PSI via LHCA3 and LHCA5. Overall, the work is very well written and informative. It is also very clear in the description of the results and precise in the discussion section where all the relevant literature has been comprehensively cited. I have a few comments that in my opinion should be addressed though, as follows:
Line 15: in my opinion the word “controlled” does not fit the meaning of the sentence.
The word “controlled” was removed and the corresponding sentence was combined with the subsequent sentence.
Lines 20-21: in this work the authors didn’t demonstrate the role of LHCSR3 in the regulation of the excitation energy transfer to PSI. I suggest toning down the sentence.
The sentence was toned down to “imply that LHCSR3 might be involved in the regulation of excitation energy transfer to the PSI core via LHCA5/LHCA3.”
Lines 82-83: I wouldn’t refer to luminal acidification as a process of acclimation. In my opinion, lumenal acidification is faster than acclimation, which calls also for proteins and pigments synthesis indeed.
The word “acclimation” was removed and the corresponding sentence was combined with the subsequent sentence.
Line 93: data within this work suggest the existence of specific protein-protein interactions but this work didn’t demonstrate their existence.
The corresponding sentence was changed to “We detected crosslinks suggesting the binding of LHCBM trimers to the LHCA1-PSAG side of PSI as well as protein-protein interactions of LHCSR3 with LHCA5 and LHCA3.
Lines 101-102: what’s the effect of crosslinking on the biological sample? Can it affect protein integrity or generate technical biases? I think this topic deserves a few sentences in the discussion section.
Treatment of isolated thylakoids with DSS did not change the solubilization behavior and migration pattern of photosynthetic protein complexes upon sucrose density gradient centrifugation (data not shown). Therefore, we are confident that crosslinking does not impact the intermolecular interactions of light harvesting antenna protein complexes, which were the focus of the presented work.
Lines 115-116: How many peptides are needed before an interaction can be claimed with sufficient confidence?
Generally, each significantly identified crosslinked peptide combination reflects a potential protein-protein interaction. Choosing a minimum number of peptides would be arbitrary. The fact that we observed crosslinking of the same LHCA1 peptide with peptides of several different LHCBM proteins substantiates the potential binding of LHCBM trimers to the LHCA1-PSAG side of PSI. This aspect was clarified in the results section.
Lines 176-177: How many algorithms are needed before an interaction can be claimed with sufficient confidence?
One algorithm is sufficient to confidently identify a potential protein-protein interaction. Each algorithm implements a different approach to identify crosslinked peptides, leading to limited overlap of results between different algorithms. We chose to use several algorithms for the identification of crosslinked peptides to maximize the number of identifications and emphasized those crosslinked peptide combinations that were identified by at least two algorithms (with the exception of the potential interaction between LHCA3 and LHCSR3, where we relied on the identification of two distinct crosslinks between the same peptides). These aspects were clarified in the results section.
Lines 214-215: I suggest being more specific in the description of the environmental conditions.
More details were added to the description of the environmental conditions in the materials and methods section as well as the results sections.
Lines 313-316: What’s the expected impact of such light environmental conditions during the growth of the biological material on the conclusions drawn within this work? I would suggest including a brief discussion on the topic.
Since identifying potential interaction partners of LHCSR3 was of interest in this work, these environmental conditions were chosen to induce expression of LHCSR3. In addition, these environmental conditions induced state transitions, which facilitated the identification of potential interactions between LHCI and LHCII. These aspects were added to the introduction and the results section.
Lines 371-375: Crosslinking was performed after thylakoids isolation. Is this practice expected to affect the reliability of the results?
Crosslinking was performed on the level of isolated thylakoids to reduce the complexity of the sample. As detailed above, this procedure is not expected to affect the reliability of the results.
Lines 382-384: It seems the thylakoids of the mutant were treated differently. Is that so? Why? This needs clarification.
Solubilization of thylakoids isolated from the lhca5 insertional mutant was performed exactly as for the corresponding control, which was the LHCA5 complemented strain. The solubilization procedure for the crosslinked sample described in the preceding sentence slightly differed due to the prior crosslinking step – but again crosslinked sample and corresponding control were treated equally.
Reviewer 2 Report
Comments and Suggestions for Authors
1. Cannot be abbreviated "Chlamydomonas reinhardtii" in the title.
2. Abstract is no charming and some important information should be added.
3. No hypothesis could be found.
4. Some references appeared in the results, it is not the normative format.
5. This manuscript is still crude and need to be polished.
Comments on the Quality of English Language
It is OK
Author Response
Thank you for your valuable suggestions.
- Cannot be abbreviated "Chlamydomonas reinhardtii" in the title.
Response: The abbreviation was removed.
- Abstract is no charming and some important information should be added.
Response: We revised the abstract to make it more concise and interesting.
- No hypothesis could be found.
Response: This work relied on a discovery-based approach. The goal was structurally determine binding of LHCBM and LHCSR proteins to PSI, which was successful.
- Some references appeared in the results, it is not the normative format.
Response: To our knowledge, it is not uncommon to include references in the results section.
- This manuscript is still crude and need to be polished.
Response: In the revision we aimed to polish the manuscript and make it more concise. It was also our goal to highlight our findings in a more comprehensive way.